# A CT-Based Radiomic Signature for the Differentiation of Pulmonary Hamartomas from Carcinoid Tumors

**DOI:** 10.3390/diagnostics12020416

**Published:** 2022-02-05

**Authors:** Ayten Kayi Cangir, Kaan Orhan, Yusuf Kahya, Ayse Uğurum Yücemen, İslam Aktürk, Hilal Ozakinci, Aysegul Gursoy Coruh, Serpil Dizbay Sak

**Affiliations:** 1Department of Thoracic Surgery Ankara, Ankara University Faculty of Medicine (AUFM), Ankara 06100, Turkey; cangir@medicine.ankara.edu.tr (A.K.C.); kahya@medicine.ankara.edu.tr (Y.K.); yucemen@medicine.ankara.edu.tr (A.U.Y.); akturk@medicine.ankara.edu.tr (İ.A.); 2Medical Design Application and Research Center (MEDITAM), Ankara University, Ankara 06100, Turkey; 3Department of Dental and Maxillofacial Radiodiagnostics, Medical University of Lublin, 20-093 Lublin, Poland; 4Department of Dentomaxillofacial Radiology, Ankara University Faculty of Dentistry, Ankara 06100, Turkey; 5Department of Pathology, Ankara University Faculty of Medicine (AUFM), Ankara 06100, Turkey; ozakinci@medicine.ankara.edu.tr (H.O.); sak@medicine.ankara.edu.tr (S.D.S.); 6Department of Radiology, Ankara University Faculty of Medicine (AUFM), Ankara 06100, Turkey; coruh@medicine.ankara.edu.tr

**Keywords:** radiomics, machine learning, pulmonary hamartomas, carcinoid

## Abstract

Radiomics is a new image processing technology developed in recent years. In this study, CT radiomic features are evaluated to differentiate pulmonary hamartomas (PHs) from pulmonary carcinoid tumors (PCTs). A total of 138 patients (78 PCTs and 60 PHs) were evaluated. The Radcloud platform (Huiying Medical Technology Co., Ltd., Beijing, China) was used for managing the data, clinical data, and subsequent radiomics analysis. Two hand-crafted radiomics models are prepared in this study: the first model includes the data regarding all of the patients to differentiate between the groups; the second model includes 78 PCTs and 38 PHs without signs of fat tissue. The separation of the training and validation datasets was performed randomly using an (8:2) ratio and 620 random seeds. The results revealed that the MLP method (RF) was best for PH (AUC = 0.999) and PCT (AUC = 0.999) for the first model (AUC = 0.836), and PC (AUC = 0.836) in the test set for the second model. Radiomics tumor features derived from CT images are useful to differentiate the carcinoid tumors from hamartomas with high accuracy. Radiomics features may be used to differentiate PHs from PCTs with high levels of accuracy, even without the presence of fat on the CT. Advances in knowledge: CT-based radiomic holds great promise for a more accurate preoperative diagnosis of solitary pulmonary nodules (SPNs).

## 1. Introduction

Pulmonary hamartomas (PHs) are the most frequently occurring benign lung tumors, which are predominantly composed of hyaline cartilage, intermixed with other mesenchymal components, including fat, smooth muscle, and bone with clefts of the entrapped respiratory epithelium [1,2]. More than 90% of PHs are peripheral, while 10% or under are endobronchial. Peripheral tumors comprise 7–14% of the total amount of radiographic solitary pulmonary nodules (SPNs) [3]. PHs are often incidental findings on imaging and can mimic pulmonary malignancies. Patients with PHs do not need any additional treatment, apart from cases in which there is the rapid growth of the tumor or the patient starts to show medical symptoms. The “popcorn” or “comma-shaped” appearance of calcification or the presence of adipose tissue is pathognomonic for PH; however, these findings are absent on CT in 30% of PHs [4]. If fat is not visible or there is no specific popcorn calcification, diagnosing PH through CT can be challenging. In particular, it is difficult to differentiate a PH from a pulmonary carcinoid tumor (PCT) if there is no fat in the lesion, so further diagnostic methods are required. Positron emission tomography/computed tomography (PET/CT) is reportedly useful for distinguishing PHs with low metabolic activity from malignancies with high metabolic activity. However, low-grade malignancies, such as PCTs, may also have low metabolic activity on PET/CT and vice versa; some PHs may reveal metabolic activity on PET/CT [4].

PCTs are malignant epithelial neuroendocrine tumors and they are morphologically and prognostically heterogeneous. Typical carcinoids are ≥0.5 cm and have <2 mitoses per 2 mm^2^ without necrosis. Atypical carcinoids have 2–10 mitoses per 2 mm^2^ and/or foci of necrosis. The CT imaging features of peripheral PCTs are round or ovoid-shaped peripheral lung nodules with smooth or lobular margins [5,6]. When the tumor is located peripherally, it is difficult to differentiate a PCT from a benign nodule, such as hamartomas, granulomas, and intrapulmonary lymph nodes. PCTs are usually highly vascular on CT, and typically increase enhancement following the administration of an intravenous contrast agent. However, not all carcinoids enhance and enhancement alone does not allow bronchial carcinoids to be differentiated from other nodules. Calcification or an endobronchial component associated with the nodule can suggest the diagnosis; however, these findings may not be useful for distinguishing PCTs from benign nodules. Unfortunately, PET-CT also has no diagnostic criteria for PCT [7]. Therefore, these patients exhibit a “lung mass” or “SPN” as opposed to a preoperative diagnosis of PCT, but the treatment for PCTs is surgical and requires more clear information. PHs and PCTs can be easily distinguished histopathologically, so a transthoracic fine-needle aspiration biopsy may be a useful method. However, PHs can be mistaken for adenocarcinomas on fine-needle aspiration biopsies if the invaginated epithelium is prominent and reactive. As a result, particularly in the cases with no adipose component, or those lacking the characteristic calcification pattern, the diagnosis of PH can be problematic and surgery becomes mandatory to rule out an underlying malignancy [8,9].

The field of radiomics is an emerging area in which imaging data is converted into a high-dimensional mineable feature space through the use of multiple automatically extracted data-characterization algorithms. Different solid tumors have different biological bases that vary due to the density of tumor proliferation and the tissue components. This heterogeneity is reflected in the calculations of the complex distribution of CT attenuation, called imaging heterogeneity. Thus, radiomics platforms are employed to manage both the imaging and clinical data as well as following radiomics statistical analysis, and they have shown promise in terms of revealing specific algorithms that can be used for quantifying a disease condition, thus providing valuable data that can be used for precision medicine [10,11,12]. 

To date, no non-invasive method has been developed to distinguish between PHs and PCTs, which have very different treatment strategies. Radiomics can be used for quantifying the features of lesions and also to possibly enhance the process of diagnosing the disease. Therefore, this research employed imaging quantification and machine learning to discriminate PHs and non-fat-tissue PHs (NFT-PH) from PCTs.

## 2. Materials and Methods

### 2.1. Patients and Dataset Management

Approval for the study protocol was obtained from the Institutional Review Board of Ankara University, the Faculty of Medicine (UFM), (IRB no: 2021-104). Due to the retrospective nature of the study, informed consent from the patients was not required. In total, surgical resection was performed on 227 patients (130 PCT and 97 PH) between 2012 and 2019, in the Thoracic Surgery Department, UFM. The inclusion criteria were: (1) surgical treatment; (2) histopathologically-confirmed PCT or hamartoma; (3) presence of CT images in the Radiology Information System/Picture Archiving and Communication System (RIS/PACS; Centricity 5.0RIS-i, GE Healthcare, Milwaukee, WI, USA) at the UFM; (4) presence of contrast-enhanced CT conducted within four weeks before the operation; (5) absence of pathognomonic calcification pattern in patients in the PH group; (6) peripheral localization of the PCT or PH on the CT scan; and (7) absence of another malignancy. A total of 138 (78 PCT and 60 PH) patients who satisfied the criteria of inclusion were enrolled in the study. The diagnosis of fat in the nodule was made if the measurement of the region of interest (ROI) was between −40 and −120 HU. No visible fat tissue was detected in 38 of 60 PHs, and the density measurements of the nodules by ROI were above −40 HU, which could be significant on their CT image in the diagnosis of PH. 

#### CT Protocol and Lesion Segmentation

Contrast-enhanced CT was conducted on each of the patients to evaluate suspected lung tumors. Either a 320-row detector CT (Toshiba Aquilion ONE, Otawara-shi, Japan), 64-row detector CT (Toshiba Aquilion 64), or 16-row detector CT (Siemens Somatom Sensation16, Forcheim, Germany) were used for performing the chest CTs. The acquisition parameters were 0.5 mm, 0.5 mm, or 0.625 mm detector collimation; 120 kVp tube voltage; 0.5 s gantry rotation time; 1 mm, 1 mm, or 1.5 mm reconstructed section thicknesses; and 0.8 mm, 0.8 mm, and 1 mm reconstruction intervals. Before conducting the examinations, the patients were injected with 60–100 mL (1–1.5 mL/kg) of nonionic intravenous contrast agent (350/100 Omnipaque, GE Healthcare, Oslo, Norway). A workstation was used to analyze the multiplanar reformatted images (GE Healthcare, Waukesha, WI, USA) (Figure 1).

The Radcloud platform (Huiying Medical Technology Co., Ltd., Beijing, China) was used for managing the imaging and clinical data as well as the following radiomics statistical analysis. The separation of the data to be used for training and validation purposes was performed randomly based on a 2:8 ratio with 620 random seeds.

Two radiomics hand-crafted models were prepared in this study: (1) the first model included all the data regarding the 138 patients to differentiate between the 78 PCTs and 60 PH; (2) the second model included 78 PCTs and 38 PHs without signs of fat tissue.

### 2.2. Image Segmentation

All images were reviewed by two senior observers (YK and UY), whose experience in the field was 10 and 5 years, respectively. The clinical information of the patients remained blinded from them and was manually delineated for all lesions. The senior clinicians reviewed all contours (KO, AGÇ) (Figure 1). In the cases where the discrepancy was ≥5%, the borders of the tumor were delineated by the senior clinicians [11]. In total, 116 Volume of Interests (VOIs) were segmented from 115 scans and used for analyses.

### 2.3. Feature Extraction 

The feature extraction in this study proposes an investigation of shape-based attributes and texture-based ones, to classify the pulmonary hamartomas and carcinoid tumors findings. To achieve a good classification, it was necessary to use effective segmentation, extract relevant attributes, and use machine learning algorithms. In the methodology the set of shape attributes is important data to the classification of findings, thus, mainly shape-based attributes, but also size-based and textural features were planned for use in this study. 

Feature extraction from the image utilizing the Radcloud platform produced via CT yielded 107 quantitative features overall, which were subsequently classified into 1 of 3 groups: Group 1 (first-order statistics) comprised 18 descriptors that provided a quantitative delineation of how the voxel intensities were distributed in the CT image by applying frequently utilized and simple metrics. Group 2 (shape- and size-based features) included 3-dimensional features that denoted the region’s size and shape. Lastly, Group 3 (texture features) included 75 textual features through which the heterogeneity differences within the region could be quantified, where the gray level run-length and gray level co-occurrence texture matrices were used for calculation. 

### 2.4. Feature Qualification

Different techniques were used for selecting the features, including the variance threshold (variance threshold = 0.8), SelectKBest, as well as the least absolute shrinkage and selection operator (LASSO) to reduce any redundant features. A threshold of 0.8 was used for the variance threshold technique, which led to the removal of eigenvalues of variances < 0.8. The SelectKBest technique, in which the single feature variables are selected, employs a *p*-value for the analysis of how the features and classification results are correlated (Figure 2) This study used all features whose *p*-value was <0.05. In the LASSO model, the L1 regularizer was utilized as the cross-function, where the value of the cross-validation error was 5 and a maximum of 1000 iterations was used (Figure 3). 

In sum, we handcrafted 2 machine learning models using the extracted features. The first model included all patients for differentiating between the PHs and PCTS from CT images as well as clinical characteristics of the patients, while the second model included all PCTs and only 38 hamartoma patients without signs of fat tissue. Same machine learning classifiers were used in this study as stated below. 

### 2.5. Statistical Analysis

The Radcloud platform was used for performing the statistical analyses. This study constructed five classifiers that were used for constructing radiomics-based models: logistic regression (LR), random forest (RF), extreme gradient boosting (XGBoost), support vector machine (SVM), and k-nearest neighbor (KNN). The validation approach was employed to enhance the model’s effectiveness. 

The parameters applied were as follows: N_neighbors (5) and weights (uniform) were applied for KNN. Kernel (rbf), C (1), gamma (auto), class_weight (balanced), decision_function_shape (ovr), and random_state was applied for SVM. Eta (0.3) and max_depth (6) were applied for XGBoost. N_estimators (10) and class_weight (none) were applied for RF. Penalty (L2), C (1), solver (liblinear), class_weight (none), multi_class (ovr), random_state, and splitter (best) were applied for LR. The gini criterion was applied for DT. 

A receiver operating characteristic (ROC) curve and area under the curve (AUC) were employed for assessing the predictive ability of the training and validation datasets, respectively. In this study, the classifier performance was evaluated according to four indicators: FP (precision = true positives/(true positives + false positives)), R (recall = rue positives/(true positives + false negatives)), F1 score (F1 score = P × R × 2/(P + R)), and support (total number in test set).

The comparison between the groups was made using the Student’s *t*-test as well as the Mann–Whitney U test. A *p*-value < 0.05 was regarded as being statistically significant.

## 3. Results

Gender, age, lesion location, and the mean diameter (±mm) distributions were compared between the PH and PCT groups, and their *p*-values were 0.03, 0.06, 0.412, and 0.58, respectively (Table 1).

Of the 107 identified features, 46 were selected for the models using the variance threshold method (Figure 1). From there, we used the best K method to select 11 features (Figure 2) and finally selected 8 optimal features using the LASSO algorithm (Figure 3). The first model included all patients. In Figure 4, the ROC curve analysis for both training and test datasets is shown for differentiating between PHs and PCTS. The AUC of the XGBoost and RF machine learning techniques had a peak value of 0.996–1 for the training dataset, whereas for the test dataset, the RF was the highest. In Table 2, the outcomes for the machine learning classifier for the test data are shown. The RF score for PH (AUC = 0.999) and PCT (AUC = 0.999) was the best method for the test set. Table 3 presents the diagnostic effectiveness utilizing the four different indicators. The PHs range between (0.87–1) for precision, (0.69–0.97) for recall, (0.80–0.98) for the F1 score, and (62) for support, whereas the PCTs ranges between (0.59–0.94) for precision, (0.77–1) for recall, (0.68–0.97) for the F1 score, and (30) for support. The RF machine learning technique yielded the highest values. 

The second model included all PCTs and only 38 hamartoma patients without signs of fat tissue. In Figure 5, the outcomes of the ROC curve for the training and test data to differentiate NFT-PHs from PCTs are presented. The AUC of the XGBoost and RF machine learning techniques had the highest values of 0.995–0.997 for the training set, whereas XGBoost and RF had the highest values of 0.820–0.836 for the test set. In Table 4, the outcomes for the machine learning classifiers for the test data are presented. The RF score for NFT-PH (AUC = 0.836) and PCTs (AUC = 0.836) indicated that it was the best method for the test set. Table 5 shows the diagnostic effectiveness of each of the specified indicators. The PHs ranged between (0.57–0.93) for precision, (0.66–0.97) for recall, (0.66–0.97) for the F1 score, and (29) for support, whereas the PCTs ranged between (0.81–0.98) for precision, (0.68–0.97) for recall, (0.75–0.98) for the F1 score, and (63) for support. The RF machine learning technique yielded the highest value. 

It was also found that eight radiomics features could differentiate between PCTS as PHs, namely the maximum two-dimensional diameter (row); maximum two-dimensional diameter (column); Long Run Low Gray Level Emphasis (LRLGLE); the dependence variance (DV); kurtosis; Large Dependence Low Gray Level Emphasis (LDLGLE); and maximum three-dimensional diameter (Figure 3).

Table 6 presents the values for the confusion matrix for PHs and PCTs using the highest learning MLP classifier (RF).

## 4. Discussion

Our results revealed that radiomics can be helpful to differentiate the PHs from PCTs. Two hand-crafted radiomics models were prepared in this study: with and without PH fat signs. The results revealed that the MLP method (RF) was best for the PH (AUC = 0.999) and PCT (AUC = 0.999) for the first model (AUC = 0.836) and PCT (AUC = 0.836) in the test set for the second model. Moreover, we defined eight radiomics features that could differentiate between the PCTs and PHs.

Radiomics involves using high-dimensional quantitative features extracted from imaging data to non-invasively quantify pathology. Recent studies have shown the potential for the application of radiomics in the oncological field [8,12]. This technique could complement the conventional approaches for analyzing the images and facilitate the process of delivering treatment tailored to individual patients [13]. There has been a recent increase in the volume of research on the applications of thoracic tumors. [14]. It is possible to extract multiple quantitative features from medical images, including CT and MRI, through the application of high-throughput computing [15]. These features include the use of intensity, shape, texture, wavelet, and LOG features to build predictive or prognostic non-invasive biomarkers for imaging modalities [8,9,10,11,12,13,14,15,16]. 

PHs do not require additional treatment other than in situations where there is the rapid growth of the tumor need no further treatment unless the lesion grows rapidly or the patient exhibits clinical symptoms [17,18]. However, surgical resection is the mainstay for PCTs, even if they are single and small. Due to these differences in treatment modalities, it is important to differentiate between PHs and PCTs using a clinical diagnostic modality. However, precise radiological differentiation criteria do not exist. The presence of fat and calcification on CT is a good indicator of a PH, but about 35% of hamartomas lack fat or calcification [17,18,19]. Although PHs can be diagnosed with an accuracy of approximately 62% by CT and 81% by PET, approximately 20% of patients may have features suggestive of malignancy [4,19,20]. Differentiating NFT-PHs from PCTs with non-invasive methods remains problematic, and the patients with an NFT-PH may undergo unnecessary surgery or needle biopsy for diagnostic and therapeutic purposes. Although it is quite easy to diagnose these tumors on tissue sections, PHs can be mistaken for adenocarcinoma in fine-needle aspiration biopsy specimens if invaginated epithelium is prominent and/or showing reactive atypia [8,17,18,19,20]. On the other hand, PCTs are usually enhanced on a CT scan after contrast administration, but this radiological finding is not diagnostic. Dense ossification, calcification, or an endobronchial component associated with the nodule suggests the diagnosis. However, a CT scan may not be useful for distinguishing PCTs from benign nodules [5]. Several researchers have previously utilized radiomics models to pathologically differentiate PHs from other pulmonary pathologies. Guan et al. differentiated PH from adenocarcinoma and found that the average contrast, cluster prominence, cluster shade, energy, and entropy were considerably higher in PHs in comparison to adenocarcinomas [8]. Another research group found that the internal structure of malignant lung tumors has greater complexity and inhomogeneity in comparison to lesions that are benign as a result of quantification via radiomics analysis [21].

To the best of the authors’ understanding, no non-invasive technique has been developed to differentiate PH/NFT-PHs from PCTs [2,4,5,6,7,17,18,19,20]. Therefore, in this study, the radiomics features extracted from CT images were used to differentiate PHs from PCTs. Since solid tumors are heterogeneous, intra-tumor heterogeneity can be determined by calculating the complex distribution of CT attenuation; this is termed imaging heterogeneity [22]. Radiomics can quantify the high-dimensional mineable features and identify underlying differences, offering a virtually unbounded stock of imaging biomarkers that have the potential to enhance diagnostic performance [21,22,23,24]. In particular, there has been a widespread application of tissue-based features in the differential diagnosis of SPN [8,14,23,24].

Studies have adopted machine learning approaches, such as segmentation, clustering, artificial neural network, Supporter Vector Machine (SVM), and Convolution Neural Network (CNN) [25,26,27,28]. CNN is originated from the functions of neurons. CNN has become the popular deep-learning model for the field of the medical imaging area. From deep learning AI algorithms, the detection of nodules was also designed [28]. In a study by Chung et al. [29], 150 CT scans (100 benign and 50 malignant cases) were included for evaluation deep learning and found that a mean AUC was 0.86 (0.81–0.91). In another recent study, Yang et al. proposed a preoperative staging tool for stage I and stage II thymoma patients based on CT images. They utilized the 3D-DenseNet model, which is an ANN deep learning model, to differentiate between MK stage I and stage II thymomas. They concluded that the ability to classify the stage of the thymoma could ultimately be used as a guide to determine surgical treatment and enhance patient outcomes. Deep learning methods can also demonstrate performance in many applications, including object, face, and activity recognition, tracking, and three-dimensional mapping, especially lung nodule detection [30].

There has been minimal utilization of machine learning systems as an artificial intelligence strategy by applying radiomics features in PHs and PCTs. Therefore, this study involved the development of a radiomics-based model with machine learning to differentiate PHs from PCTs. Similar to the findings of Guan et al. [8], the texture-GLCM analysis yielded the highest efficacy in discriminating PHs from PCTs. Specifically, the RF classifier trained with texture-GLCM features performed significantly better in correctly diagnosing PH patients than the other MLP classifiers for both models, and it can be used clinically to distinguish PHs from PCTs. This kind of machine-learning model in which texture-GLCM features are used for training, facilitate the identification of the internal distribution of attenuation and offer evidence for discrimination. The results of this study suggest that a good agreement can be achieved using radiomics features, even without the presence of fat; however, fat is still very important for differentiating PCTs from PHs. Given the results of the study, it can be recommended that RF machine learning with the presence of fat on the CT image’s feature extraction can be best used for the differentiation of pulmonary hamartomas from carcinoid tumors. 

The study contains certain limitations that should be addressed. Firstly, the data of patients from only one center were reviewed retrospectively, thus potentially causing selection bias. Secondly, the sample size was small. Larger multicenter studies with larger sample sizes will be required to validate and expand on our results. Third, ROI segmentation was performed manually, which may have been affected by subjective bias. Last, the raw images were collected from different CT scanners. Variability in image acquisition could have influenced the results, however, this is inevitable in clinical practice. 

This is the first study to distinguish PCTs from PHs with or without the presence of fat using CT image radiomics features. This technique holds great promise for a more accurate preoperative diagnosis of SPNs.

## 5. Conclusions

The study findings indicate that predictive models, including radiomics tumor features derived from CT images, are useful to differentiate pulmonary carcinoid tumors from hamartomas with high accuracy, even without the presence of fat on the CT image. By using radiomics, which is a non-invasive method, patients with PH will be saved from surgery that hast the possibility of morbidity and mortality by distinguishing pulmonary hamartoma from pulmonary carcinoids, and unnecessary economic losses will be prevented. In addition, considering the results of this study, it may be possible for patients with PC, a pulmonary tumor, to reach the right treatment in a short time with radiomics applications in the future.

## Figures and Tables

**Figure 1 diagnostics-12-00416-f001:**
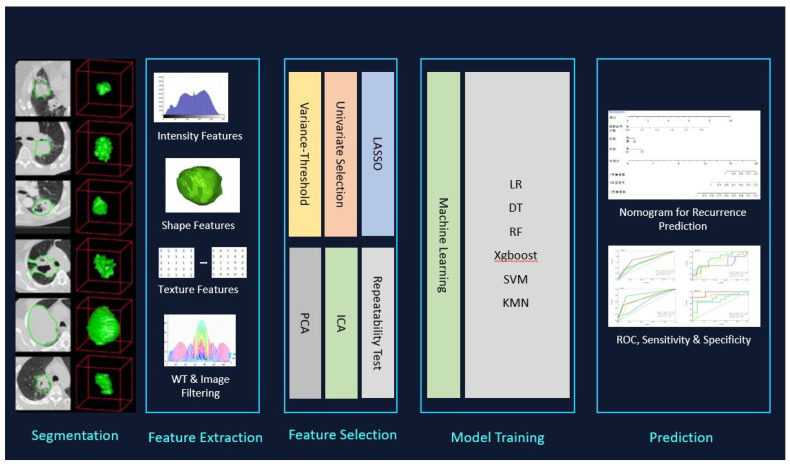
3D segmentation, feature selection, and radiomics analysis of workflow used for PCTs and PHs.

**Figure 2 diagnostics-12-00416-f002:**
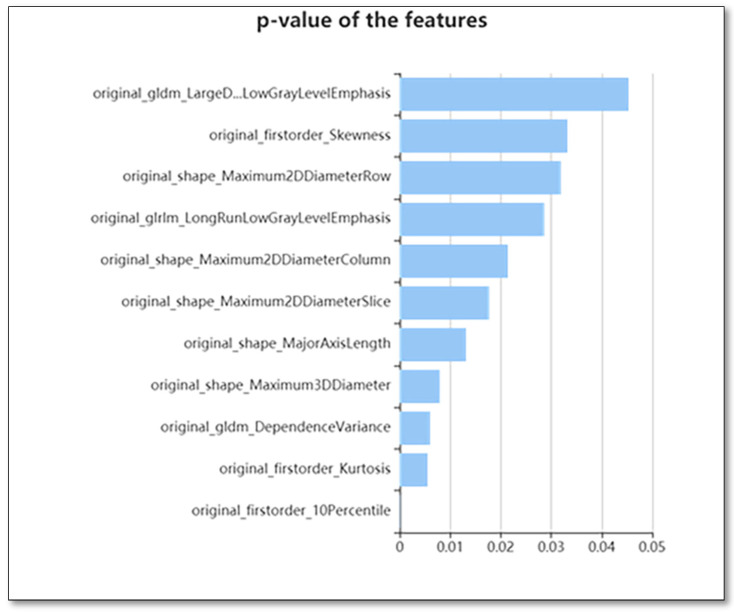
The use of K best on the feature selection to further select radiomics features, which results in 11 features.

**Figure 3 diagnostics-12-00416-f003:**
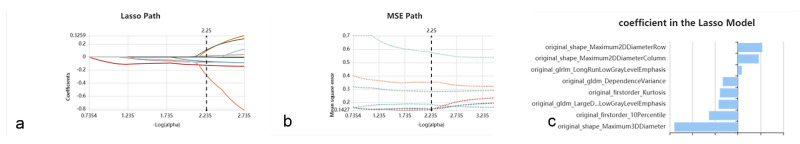
Lasso algorithm on feature selection. (**a**) Lasso path; (**b**) MSE path; and (**c**) coefficients in Lass model. Using the Lasso model, eight optimal features that correspond to the optimal alpha value were selected.

**Figure 4 diagnostics-12-00416-f004:**
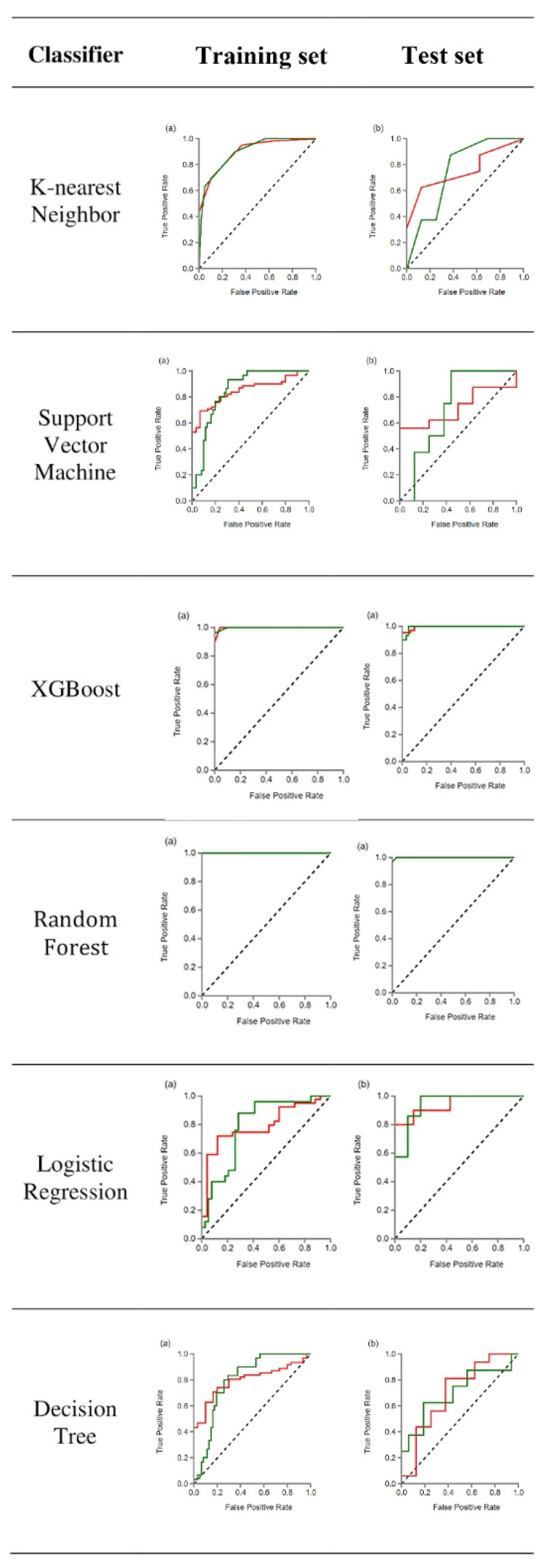
The figure shows the results of the 1st modeling ROC curve analysis for the training and test data for differentiating between PHs and PCTs. Note that green—PHs and red—PCTs. (**a**) ROC curve of the training set and (**b**) ROC curve test set.

**Figure 5 diagnostics-12-00416-f005:**
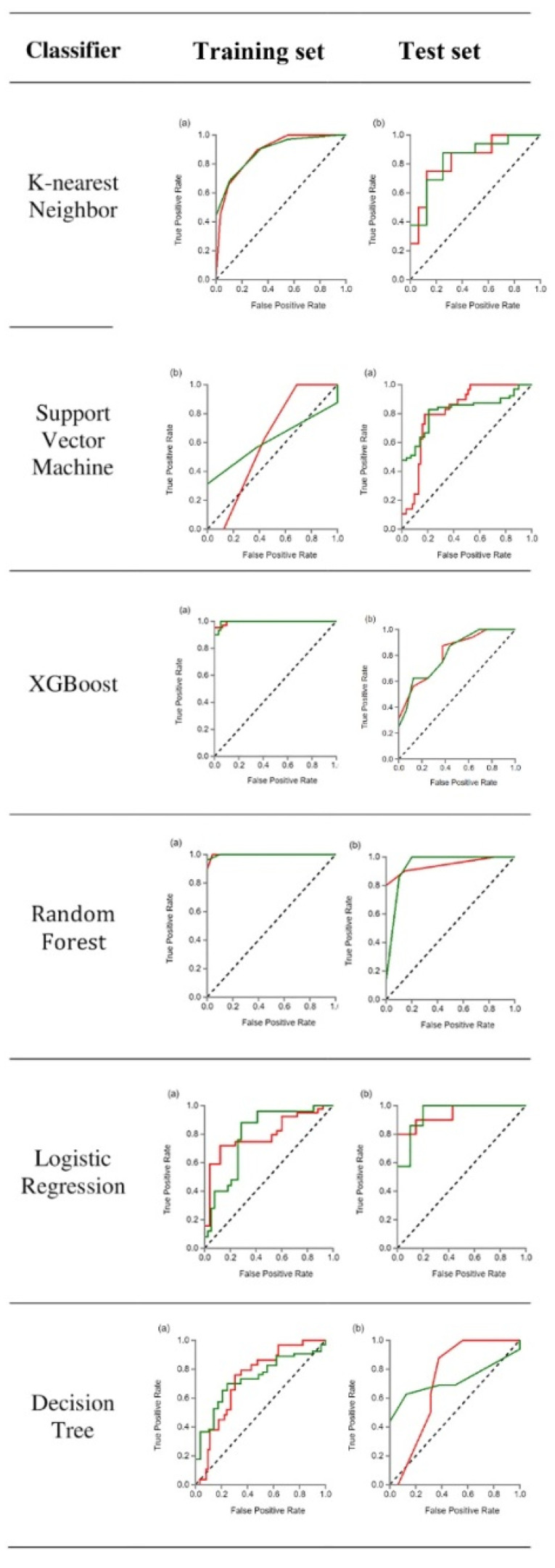
The figure shows the results of the 2nd modeling ROC curve analysis results for the training and test data for differentiating between PHs and PCTs. Note that green—PHs and red—PCTs. (**a**) ROC curve of the training set and (**b**) ROC curve of the test set.

**Table 1 diagnostics-12-00416-t001:** Clinical characteristics of the patients.

Characteristics	Pulmonary Carcinoid Tumor, (n = 78)	Pulmonary Hamartoma, (n = 60)	*p*-Value
Male	35	38	
Female	43	22	0.03
Age, median (range) (years)	52 (20–81)	55 (27–72)	0.06
Tumor site, n			
Right lung	44	38	0.412
Left lung	34	22	
Tumor diameter, mean (range) (mm) on CT	26.2 (8–70)	19.8 (8–120)	0.58

**Table 2 diagnostics-12-00416-t002:** ROC outcomes with six machine learning classifiers for the test set using the first model.

Classifiers	Category	AUC	95% CI	Sensitivity	Specificity
KNN	PHs	0.898	0.82–0.98	0.69	0.9
	PCTs	0.898	0.82–0.98	0.90	0.69
SVM	PHs	0.849	0.76–0.94	0.76	0.77
	PCTs	0.849	0.76–0.94	0.77	0.76
XGBoost	PHs	0.996	0.95–1.00	0.95	0.93
	PCTs	0.996	0.95–1.00	0.93	0.95
RF	PHs	0.999	0.98–1.00	0.97	1
	PCTs	0.999	0.98–1.00	1.00	0.97
LR	PHs	0.809	0.71–0.90	0.73	0.73
	PCTs	0.809	0.71–0.90	0.73	0.73
DT	PHs	0.806	0.73–0.90	0.75	0.75
	PCTs	0.806	0.73–0.90	0.75	0.75

**Table 3 diagnostics-12-00416-t003:** Results of the four indicators of precision, recall, F1 score, and support based on the test data using the first model.

	Indicators	KNN	SVM	XGBoost	RF	LR	DT
PHs	Precision	0.93	0.87	0.97	1.00	0.85	0.75
	Recall	0.69	0.76	0.95	0.97	0.73	075
	F1 score	0.80	0.81	0.96	0.98	0.78	0.75
	Support	62.00	62.00	62.00	62.00	62.00	62.00
PCTs	Precision	0.59	0.61	0.90	0.94	0.56	0.56
	Recall	0.90	0.77	0.93	1.00	0.73	0.77
	F1 score	0.71	0.68	0.92	0.97	0.64	0.76
	Support	30.00	30.00	30.00	30.00	30.00	30.00

**Table 4 diagnostics-12-00416-t004:** ROC outcomes for six machine learning classifiers of the test set and the second model.

Classifiers	Category	AUC	95% CI	Sensitivity	Specificity
KNN	PHs	0.613	0.39–0.84	0.62	0.56
	PCTs	0.613	0.39–0.84	0.56	0.63
SVM	PHs	0.676	0.45–0.90	0.50	0.69
	PCTs	0.676	0.45–0.90	0.69	0.5
XGBoost	PHs	0.82	0.66–0.98	0.88	0.81
	PCTs	0.82	0.66–0.98	0.81	0.88
RF	PHs	0.836	0.66–1.00	0.88	0.69
	PCTs	0.836	0.66–1.00	0.69	0.88
LR	PHs	0.723	0.55–0.90	0.88	0.63
	PCTs	0.723	0.55–0.90	0.62	0.88
DT	PHs	0.563	0.35 0.78	0.38	0.75
	PCTs	0.563	0.35–0.78	0.75	0.38

**Table 5 diagnostics-12-00416-t005:** Outcomes for four indicators, including precision, recall, F1 score, and support for the test set using the second model.

	Indicators	KNN	SVM	XGBoost	RF	LR	DT
PHs	Precision	0.57	0.68	0.93	0.93	0.50	1.00
	Recall	0.90	0.79	0.97	0.97	0.66	1.00
	F1 score	0.69	0.73	0.95	0.95	0.57	1.00
	Support	29	29	29	29	29	29
PCTs	Precision	0.93	0.90	0.98	0.98	0.81	1.00
	Recall	0.68	0.83	0.97	0.97	0.70	1.00
	F1 score	0.79	0.86	0.98	0.98	0.75	1.00
	Support	63	63	63	63	63	63

**Table 6 diagnostics-12-00416-t006:** Details of the confusion matrix for PHs and PCTs using the highest learning MLP classifier (RF) for all patients in the first and second models.

Types of Pathology	RF (1st Modeling)			RF (2nd Modeling)		
	True	False	Accuracy (%)	True	False	Accuracy (%)
PHs	12	0	100	12	4	75
PCTs	61	1	98.38	62	4	91.1
Accuracy (%)			99%			83%

## Data Availability

The data presented in this study are available on request from the corresponding author.

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
