# Peer review of "A CT-Based Radiomic Signature for the Differentiation of Pulmonary Hamartomas from Carcinoid Tumors"

_diagnostics, 2022, doi:10.3390/diagnostics12020416_

Round 1

Reviewer 1 Report

Thanks for giving me an opportunity to review this paper. In this paper, the authors evaluated CT radiomic features to differentiate pulmonary hamartomas (PH) from pulmonary carcinoid tumors (PCT). It is well-written manuscript but several issues need to be resolved before being considered for publication.

  1. In abstract: line 33-34: “Advances in ---- of SPNs”. Please write a full form of SPNs.
  2. In abstract: Line 27: “training and validation dataset was performed randomly using a 2:8 ratio”. Why training set is less and validation set is high. Usually, we use 7:3 or 8:2 as a training and validation set. You have used reversed, do you have any explanation.
  3. In introduction: line 48: “if vat….” What is vat?
  4. Line 134: , w whose… what is w?
  5. Results: line 181-182: please provide a table for comparison of two groups (PH and PCT)
  6. Please describe first and second models in the method part
  7. In discussion: Please write principle findings in the first paragraph.
  8. In discussion: Please write one paragraph for clinical implications
  9. Conclusion: it is too short. Please explain more about your study findings.
  10.  

Author Response

                                                                                                                           27/01/2022

Dear Editor;

First of all, I would like to thank you very much for your kind concern about our manuscript titled “CT-based radiomic signature for differentiation of pulmonary hamartomas from carcinoid tumors” and ID diagnostics-1560475. Secondly we carefully read your and the reviewers’ suggestions and corrected our manuscript according to them as per your suggestion. All corrections were made in MS Word program. The corrections that were made in the text are below and the changes marked as colored texts (red) in the manuscript. The corrections necessary in terms of syntax, grammar, punctuation, capitalization and spacing were made, and the entire manuscript was proofread by an expert.

Once again, thank you very much for your kind concern.

King Regards

Corresponding Author:  Prof. Dr. Kaan Orhan

Ankara University Medical Design Application and Research Center (MEDITAM), Ankara, Turkey;

Department of Dental and Maxillofacial Radiodiagnostics, Medical University of Lublin, Po   land;

Ankara University Faculty of Dentistry, Department of Dentomaxillofacial Radiology,  Ankara, Turkey; e-mail: [email protected]

The following are the changes made in the revision of the manuscript;

Reviewers'comments:

Reviewer#1

  1. In abstract: line 33-34: “Advances in ---- of SPNs”. Please write a full form of SPNs.
    We agreed with the reviewer on the issue. Full form of SPN has been added as “solitary pulmonary nodules” to the abstract section line 33-34.
  2. In abstract: Line 27: “training and validation dataset was performed randomly using a 2:8 ratio”. Why training set is less and validation set is high. Usually, we use 7:3 or 8:2 as a training and validation set. You have used reversed, do you have any explanation.

We agreed with the reviewer on the issue. There is misspelling with the written ratio (2:8). It has been corrected as “(8:2)” in the line 27.

  1. In introduction: line 48: “if vat….” What is vat?

We agreed with the reviewer on the issue. There is a misspelling with the word vat. It has been corrected as “fat” in line 48.

  1. Line 134: , w whose… what is w?

We agreed with the reviewer on the issue. There is a misspelling and “w” has been erased.

  1. Results: line 181-182: please provide a table for comparison of two groups (PH and PCT)

We agreed with the reviewer on the issue and table has been added for comparison of two groups (PH and PCT) as table 1 to the results section.

  1. Please describe first and second models in the method part.

It has been already described at the last paragraph of “2.1. Patients and dataset management section” as “Two radiomics hand-crafted models were prepared in this study: (1) the first model included all data regarding the 138 patients to differentiate between 78 PCTs and 60 PH; (2) the second model included 78 PCTs and 38 PHs without signs of fat tissue.”

But we also added in that section as well according to reviewer’s suggestion.

  1. In discussion: Please write principal findings in the first paragraph.

Added according to reviewer’s suggestion.

  1. In discussion: Please write one paragraph for clinical implications

We already wrote about clinical implications at the 2nd and 3rd paragraph of discussion section as “Dense ossification, calcification, or an endobronchial component associated with the nodule suggest the diagnosis. However, a CT scan may not be useful for distinguishing PCTs from benign nodules [5]. A number of researchers have previously utilized radiomics models to pathologically differentiate PHs from other pulmonary pa-thologies. Guan et al. differentiated PH from adenocarcinoma and found that the average contrast, cluster prominence, cluster shade, energy, and entropy were considerably higher in PHs in comparison to adenocarcinomas [8]. Another research group found that the internal structure of malignant lung tumors has greater complexity and inhomogeneity in comparison to lesions that are benign as a result of quantification via radiomics analysis [24].

To the best of the authors’ understanding, no non-invasive technique has been de-veloped to differentiate PH/NFT-PHs from PCTs [2,4,5-7,17-20]. Therefore, in this study, the radiomics features extracted from CT images were used to differentiate PHs from PCTs. Because solid tumors are heterogeneous, intra-tumor heterogeneity can be determined by calculating the complex distribution of CT attenuation; this is termed imaging heterogeneity [21]. Radiomics can quantify high-dimensional mineable features and identify underlying differences, offering a virtually unbounded stock of imaging biomarkers that have the potential to enhance the diagnostic performance [21-24]. In particular, there has been widespread application of tissue-based features in the differential diagnosis of SPN”

  1. Conclusion: it is too short. Please explain more about your study findings

We agreed with the reviewer on the issue and added sentences about study findings to the conclusion section as “By using radiomics, which is a non-invasive method, patients with PH will be saved from surgery that hast the possibility of morbidity and mortality by distinguishing of pulmonary hamartoma from pulmonary carcinoids and unnecessary economic losses will be prevented. In addition, considering the results of this study, it may be possible for patients with PC, a pulmonary tumor, to reach the right treatment in a short time with radiomic applications in the future.”

Reviewer 2 Report

This paper studies CT radiomic features evaluation to differentiate pulmonary hamartomas (PH) from pulmonary carcinoid tumors.
Overall, it is an interesting study with detailed experiments.
1. Section 2.3: It would be good to discuss the 'motivation' of these three hand-crafted features proposed in the paper, including statistics-based, shape-based, and texture-based methods.
2. Lines 94-98: There are a few '...' that should be removed.
3. Figures 2 and 3 are too small, it could be enlarged.
4. In Table 3, the results should have the same precision (i.e., two or three decimal places).
5. Section 4: Since a few methods are evaluated in this paper, what would be recommended method?
6. There are two figures in Supplementary Materials. I think they can be added to the main article.
7. How about the deep learning-based method for this research problem? It would be good to add some references in this area.

Author Response

                                                                                                                           27/01/2022

Dear Editor;

First of all, I would like to thank you very much for your kind concern about our manuscript titled “CT-based radiomic signature for differentiation of pulmonary hamartomas from carcinoid tumors” and ID diagnostics-1560475. Secondly we carefully read your and the reviewers’ suggestions and corrected our manuscript according to them as per your suggestion. All corrections were made in MS Word program. The corrections that were made in the text are below and the changes marked as colored texts (red) in the manuscript. The corrections necessary in terms of syntax, grammar, punctuation, capitalization and spacing were made, and the entire manuscript was proofread by an expert.

Once again, thank you very much for your kind concern.

King Regards

Corresponding Author:  Prof. Dr. Kaan Orhan

Ankara University Medical Design Application and Research Center (MEDITAM), Ankara, Turkey;

Department of Dental and Maxillofacial Radiodiagnostics, Medical University of Lublin, Po   land;

Ankara University Faculty of Dentistry, Department of Dentomaxillofacial Radiology,  Ankara, Turkey; e-mail: [email protected]

The following are the changes made in the revision of the manuscript;

Reviewers'comments:

Reviewer#2

  1. Section 2.3: It would be good to discuss the 'motivation' of these three hand-crafted features proposed in the paper, including statistics-based, shape-based, and texture-based methods.

Following text was added in the manuscript, “The feature extraction in this study proposes an investigation of shape-based attributes and texture-based ones to classify pulmonary hamartomas and carcinoid tumors findings. In order to achieve a good classification, it was necessary to use an effective seg-mentation, extracting relevant attributes and to use machine learning algorithms. In the methodology the set of shape attributes is an important data to classification of findings, Thus, mainly shape-based attributes but also size based and textural features were planned to use in this study.

  1. Lines 94-98: There are a few '...' that should be removed.

Corrected according to reviewer suggestions.

  1. Figures 2 and 3 are too small, it could be enlarged.

Corrected according to the reviewer suggestion. However, it should be stated that in the publication phase, the more resolution figures will be sent to the production.

  1.  In Table 3, the results should have the same precision (i.e., two or three decimal places).

We agreed with the reviewer on the issue and results rearranged as same precision (decimal places).

  1. Section 4: Since a few methods are evaluated in this paper, what would be recommended method?

Added in the text as, “ Given the results of the study, it can be recommended that RF

machine learning can be best used for differentiation of pulmonary hamartomas from

carcinoid tumors”.

  1. There are two figures in Supplementary Materials. I think they can be added to the main article.

Corrected and added according to reviewer’s suggestion. 

  1. How about the deep learning-based method for this research problem? It would be good to add some references in this area

The deep learning studies were added according to reviewer’s suggestion in discussion with additional references.  

Round 2

Reviewer 1 Report

Thanks for your revised version. The authors have addressed all of the comments. 

Reviewer 2 Report

The revision is fine. Thank you.